# Single-Cell Sequencing Identifies Master Regulators Affected by Panobinostat in Neuroblastoma Cells

**DOI:** 10.3390/genes13122240

**Published:** 2022-11-29

**Authors:** Giorgio Milazzo, Giovanni Perini, Federico M. Giorgi

**Affiliations:** Department of Pharmacy and Biotechnology, University of Bologna, Via Selmi 3, 40126 Bologna, Italy

**Keywords:** panobinostat, Kelly, neuroblastoma, gene networks, master regulator analysis, single-cell sequencing, transcriptomics

## Abstract

The molecular mechanisms and gene regulatory networks sustaining cell proliferation in neuroblastoma (NBL) cells are still not fully understood. In this tumor context, it has been proposed that anti-proliferative drugs, such as the pan-HDAC inhibitor panobinostat, could be tested to mitigate tumor progression. Here, we set out to investigate the effects of panobinostat treatment at the unprecedented resolution offered by single-cell sequencing. We identified a global senescence signature paired with reduction in proliferation in treated Kelly cells and more isolated transcriptional responses compatible with early neuronal differentiation. Using master regulator analysis, we identified BAZ1A, HCFC1, MAZ, and ZNF146 as the transcriptional regulators most significantly repressed by panobinostat. Experimental silencing of these transcription factors (TFs) confirmed their role in sustaining NBL cell proliferation in vitro.

## 1. Introduction

Neuroblastoma (NBL) is a type of childhood extracranial solid tumor [1], which accounts for 13% of all pediatric cancers [2] and has roughly 1000 new diagnoses in Europe alone [3]. MYCN genomic amplification is observed in 20–25% of all NBL cases [4], and the overexpression of MYCN and its target genes is one of the strongest predictors of poor outcome in NBL [5]. Beyond MYCN, several other master regulators of NBL have been identified over the last few decades, including MAX [6] and TFAP4 [7]. Despite the existence of several therapies against this disease, there is still a low survival rate for NBL, especially for late-stage disease (<30%), and the identification of more elements in the molecular machinery supporting NBL initiation and progression is of paramount importance for future NBL drug discovery and treatment [2,8].

Among the molecular therapies for the treatment of NBL, panobinostat (PNB) has recently surged in importance, with proven effects in the reduction of tumor growth in cell lines [9,10], animal models [11], and clinical patients [12]. PNB is a pan-inhibitor of histone deacetylases (HDACs), a family of chromatin-remodeling proteins that generally act as wide inducers of chromatin closure and transcriptional repression [13]. While its pharmacokinetic mechanism is known, the effects of PNB have still not been fully elucidated, and the drug, while approved for treatment of patients with relapsed multiple myeloma (MM) by the FDA [14], is currently undergoing a phase II clinical trial for treatment of childhood brain tumors [15].

With the aim of fully elucidating the effect of drug treatment at the cellular level, recent years have witnessed an avalanche in the usage of genome-wide and transcriptome-wide high-throughput sequencing technologies to highlight the molecular responders of such treatments [16]. Recent improvements in the resolution of high-throughput sequencing—specifically, single-cell sequencing—have provided an even higher power of detection for differential expression and analyses based on quantitative transcriptomics [17]. In parallel, efforts in bioinformatics have provided the scientific community with algorithms to hierarchically prioritize responders to drug treatment, highlighting the “master regulators” (MRs) as bottleneck transcription factors responsible for the differential expression of hundreds, if not thousands, of downstream genes [18]. Identifying these MRs constitutes not only a way to understand which parts of the gene regulatory networks are most affected by a treatment but also makes it possible to identify a small number of molecular targets that can be used to modulate and/or maximize the effects of pharmacological treatment [19,20].

In the current study, we tested the effects of PNB at the single-cell level, performing a drug treatment using Kelly cells, a well-known model of MYCN-amplified NBL [21,22]. We investigated the effects of PNB using the resolution power offered by single-cell RNA sequencing, defining the gene networks most affected by this drug at the transcriptional level. Then, using master regulator analysis (MRA), we identified the MRs most inhibited by HDAC inhibition, testing their proliferative and pro-oncogenic effects with targeted experiments in vitro.

## 2. Materials and Methods

### 2.1. Cell Culture, Treatment, and Single-Cell Sequencing

Kelly cells were cultured in RPMI 1640 (Sigma-Aldrich^®^, St. Louis, MO, USA) with 10% FBS (Gibco^®^, Waltham, MA, USA), along with 1% penicillin/streptomycin and 2 mM L-glutamine. Cells were kept in a humidified atmosphere with 5% CO_2_ at 37 °C. The medium was substituted biweekly, with passages for maintenance at 80% in the culture flasks. Kelly cells were treated with panobinostat (10 nM) and DMSO as a vehicle control for 12 h prior to single-cell sequencing. The dosage was identical to that previously used for panobinostat treatment of neuroblastoma cells [10] and similar to that normally used in the literature for cell cultures [23,24].

Cell hashing [25] was performed following the manufacturer’s instructions (TotalSeq™-A) and 10X Genomics^®^ (Pleasanton, CA, USA) sequencing was performed using standard protocols, as explained in detail in our previous work [26]. Libraries were then sequenced in 150 bp paired-end mode on an Illumina^®^ (San Diego, CA, USA) NovaSeq 6000.

### 2.2. Bioinformatics Analysis

Raw reads were aligned to the human genome version hg38 using Cellranger software version 3.1.0 (10X Genomics^®^, Pleasanton, CA, USA) based on STAR Aligner version 2.7 [27]. Raw cell-by-cell gene expression counts were loaded using the Seurat R package version 4.1.1 [28]. Cells with less than 1000 detected genes were discarded, as well as genes present in fewer than three cells. Cells were assigned to the control (hashtag 1) or panobinostat (hashtag 2) group using the following criteria: cells with no hashtag counts for one group but at least one hashtag count for the other group were assigned to the latter; otherwise, cells were assigned to a specific group if the hashtag counts for the group were at least ten times higher than the other group. All other cells were deemed “multiplets” with an ambiguous origin (or as possibly arising from a sequencing droplet incorporating two or more cells from different groups).

Data were log-normalized prior to dimensionality reduction analysis and differential expression analysis. Differential expression was calculated using the Wilcoxon test and *p*-values were adjusted using the Benjamini–Hochberg method [29].

Gene set enrichment analysis was performed using the R package *fgsea* [30] with pathways defined by MSigDB [31]. The selected pathways were as follows: for senescence, we used Fridman_Senescence_UP [32]; for proliferation, we used Peart_Hdac_Proliferation Cluster_DN (proliferative genes known to be repressed by Hdac inhibitors) [33]; Heller_HDAC_Targets_up was used for genes activated by HDAC inhibitors [34]; for neuronal differentiation, we used the Gene Ontology Biological Process (GOBP) “Neuron Differentiation” pathway.

Master regulator analysis was performed using the R package viper [35] with neuroblastoma-specific gene interaction models derived from NRC and TARGET networks [20]. Briefly, master regulator analysis involves checking, for each TF-centered weighted gene network provided, the differential expression value for each gene according to the contrast panobinostat vs. control. Each gene contribution is cumulated into a TF-specific normalized enrichment score. More technical details on the procedure are provided in the viper Bioconductor package (specifically, the msviper function).

Survival *p*-values were calculated using the Wald test [20] and integrated across datasets using the Fisher *p*-value integration method, as implemented by the *corto* package [36].

All statistical analyses and image generation were performed using R software version 4.2.1 [37].

### 2.3. Doxycycline-Inducible shRNA Expression Cell Lines

The lentiviral doxycycline-inducible GFP-IRES-shRNA FH1tUTG construct was used to generate BAZ1A, HCHFC1, MAZ, and ZNF146 shRNA-expressing constructs, as well as neuroblastoma cell lines stably expressing the constructs. All the shRNA target sequences are reported in Appendix A. Sense and antisense shRNA oligonucleotides were annealed, phosphorylated (PNK, NEB Biolabs, Ipswich, MA, USA), and cloned (T4-DNA ligase, NEB biolabs, Ipswich, MA, USA) into the doxycycline-inducible GFP-IRES-shRNA FH1tUTG lentiviral plasmid. All the doxycycline-inducible GFP-IRES-control shRNA constructs were co-transfected (Effectene, QIAGEN, Germantown, MD, USA) with packaging vectors PSPAX2 and PMD2G into HEK-293T cells to induce viral production (3 mL in a six-well format). Forty-eight hours after the transfection, lentiviral media were collected, filtered through a 0.4 µm filter, and employed to infect neuroblastoma cells with polybrene (8 µg/mL) (Santa Cruz Biotechnology, Santa Cruz, CA, USA) for 48 h. Fluorescence-activated cell sorting was performed with a BIORAD S3e Cell Sorter to select Kelly cells with high GFP protein expression. Cells were treated with 2 µg/mL doxycycline (Sigma) to induce shRNA expression or with a DMSO vehicle control otherwise.

### 2.4. Reverse Transcription and Quantitative Real-Time PCR (RT-qPCR)

RNA was extracted from cells with a TRI-REAGENT (SIGMA, St. Louis, MO, USA) and quantified with a Nanodrop spectrophotometer (Thermo Fisher Scientific, Waltham, MA, USA) according to the manufacturer’s instructions. Contaminant genomic DNA was digested using a DNA-free kit (AMBION, Waltham, MA, USA). cDNAs were then synthesized with a cDNA synthesis kit (BIORAD, Hercules, CA, USA) using a mix of oligo dT and random hexanucleotide primers. RT-PCR was performed using gene-specific primers (Appendix A) and SSOADVANCE SYBR Green Master Mix (BIORAD, Hercules, CA, USA) as the fluorescent dye in a CFX96 real-time PCR system (BIORAD, Hercules, CA, USA). All primers were synthesized by Sigma (Sigma Aldrich, St. Louis, MO, USA). Following RT-qPCR, the comparative threshold cycle (ΔΔCt) method was used to evaluate fold changes in target genes relative to the reference gene GUSB, using no-doxycycline treatment as a control sample.

## 3. Results

### 3.1. Transcriptional Effects of Panobinostat on Kelly Cells at Single-Cell Resolution

Kelly cells were treated with PNB (10 nM) for 12 h and 24 h or DMSO as a vehicle control and total protein lysates were immunoblotted with specific antibodies for histone H3 lysine 27 acetylation (H3-K27ac) or for histone H3 pan-acetylation (H3-Pan_ac). Immunoblot analysis of both acetylated markers strongly confirmed the pan-HDAC inhibition in NBL Kelly cells (Figure 1A). Next, in order to perform single-cell RNA-seq analyses, Kelly cells were treated with PNB 10 nM for 12 h.

After quality filtering (see the Section 2), we obtained full transcriptome-wide profiling of 1454 untreated cells and 1031 cells from the panobinostat-treated flask, while 193 cells had an ambiguous origin (Figure 1B). We deemed the method of separation between groups based on cell hashing reliable, as the expression of hashtags was strong compared to all other genes (Figure 1C). Among the genes, we found that those most expressed in the dataset, with low dispersion and variability between cells, were encoded by L and S ribosomal protein-encoding genes, such as RPS2 and RPL3 (Figure 1C). This was consistent with other single-cell experiments performed on neuroblastoma cell lines [21,26] and is a standard feature of most human cells [38], with the stable and abundant expression of ribosomal-encoding mRNAs guaranteeing a robust translation machinery [39]. Other highly abundant expressors in Kelly cells were well-known genes, such as the cyclooxygenase-encoding COX1, COX2, and COX3 [40] and MALAT1, transcribed into the long-coding transcript with various structural and transcriptional regulatory functions [41]. Two mitochondrially encoded genes were also amongst the most abundantly expressed; namely, the NADH dehydrogenases 1 and 4 (ND1 and ND4) and CYTB (MT-CYB), encoding for cytochrome B. The housekeeping gene GAPDH was also amongst the most expressed, with little change between cells (Figure 1C) or between panobinostat-treated and control cells (Appendix A, log2FC = 0.11), justifying its current usage as a control for quantitative experiments such as RT-PCR. MYCN, a highly amplified oncogene encoding for transcription-factor Kelly cells and a known driver of neuroblastoma progression [5], was also amongst the most abundant transcripts with a normalized expression of 14.6 (log-normalized: 2.68), placing it as the 106th most expressed gene in Kelly cells.

Data dimensionality reduction showed a clear separation between panobinostat-treated cells and untreated (control) cells based on the overall transcription proFiles, with ambiguously assigned cells constituting a separate group (Figure 1D). The panobinostat treatment was clearly the strongest source of variability and perturbation in the dataset (Figure 1D). We identified the actual abundance of reads per cell as another element of strong cell separation, with groups representing cells at different sequencing depths (Figure 1E) and two minor groups, located between the major clusters, characterized by low coverage (between 2000 and 10,000 reads/cell). Interestingly, all the most ambiguous cells were characterized by low sequencing coverage (<10,000 reads/cell). We then decided to remove all ambiguously assigned cells in all the analyses described following this paragraph, starting from gene-level differential expression analysis.

Another source of dataset variability was the cell cycle-based transcriptional signature: both treated and untreated cells showed clear subgroups associated with different phases of the cell cycle—G1, S and G2M (Figure 1F)—assigned in silico using reference-based phase assignment [42]. In this case, panobinostat seemed to induce a significant stop in cell cycle progression, as previously reported [43], with cells in the G1 phase moving from 25.39% in untreated cells to 47.65% in treated cells (Table 1).

Despite the possibility of removing the cell cycle and the abundance of reads per cell via linear regression (t-SNE projection post-removal of these sources of variability is shown in Appendix A), we decided to proceed with the next steps of the analysis without altering the original dataset, as we thought both the cell cycle phase and RNA abundance were elements of biological importance.

**Figure 1 genes-13-02240-f001:**
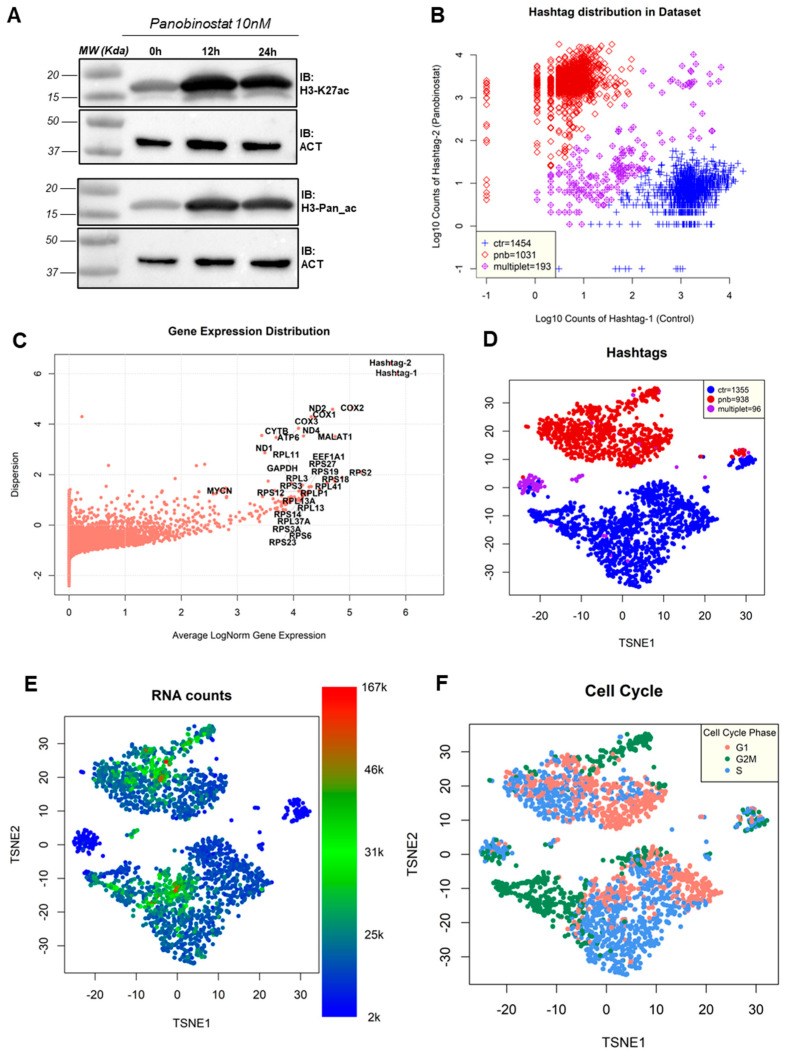
(**A**) Immunoblot analysis for histone H3 lysine 27 acetylation (H3-K27ac) or histone H3 pan-acetylation (H3-Pan_ac) upon treatment with 10 nM panobinostat at time points 0, 12, and 24 h; β-actin (ACT) was used as a loading control. (**B**) Distribution of Kelly cells according to hashtag expression. X-axis: hashtag 1 (associated with control cells), y-axis: hashtag 2 (associated with PNB-treated cells). The legend indicates the number of cells assigned to three groups: prevalently control (ctr), prevalently treated (pnb), and ambiguous (multiplet). (**C**) Distribution of genes in the experiment plotted by average log-normalized gene expression (x-axis) and by dispersion (y-axis). Labels are reported for genes (and hashtags) with the highest overall expression (MYCN is reported for comparison). (**D**) t-distributed stochastic neighbor embedding (t-SNE) projection of the dataset prior to removal of ambiguous hashtag cells; colors indicate the cell assignment: control (ctr), PNB-treated (pnb), and ambiguous (multiplet). (**E**) t-SNE projection, as in the previous panel, with colors indicating the number of reads detected in each cell. (**F**) t-SNE projection, as in the previous panel, with colors indicating the cell cycle phase predicted in silico according to known cell markers [42].

### 3.2. Effects of Panobinostat at Gene Level and Pathway Level

First, we checked whether control Kelly cells generated by our single-cell sequencing setup would indeed recapitulate the transcriptomics of a standard bulk RNA-Seq experiment. In order to do so, we combined the contributions of each of the 1454 control cells by adding the raw read counts for each gene, thus generating a transcriptome-wide gene-by-gene expression readout for the plate. We then compared these values (normalized as reads per million mapped reads (RPM)) with those from bulk RNA-Seq data for untreated Kelly cells derived from the NBL cell line RNA-Seq database [22]; the correlation between single-cell sequencing and bulk RNA-Seq was high (Spearman’s correlation coefficient (SCC) = 0.9), showing that the single-cell data could recapitulate bulk data (Figure 2A). Our results also confirmed that the common genes used as housekeeping markers for the Kelly cell line (GAPDH and ACTB) are indeed highly expressed in both bulk and single-cell data, while MYCN is the activated paralog of the MYC family in this cell type, with MYCL and MYC being detected only in trace amounts (Figure 2A). We then processed the single-cell pseudobulk samples (control and panobinostat-treated Kelly cells summed to obtain two samples) via the dimensionality reduction algorithm t-SNE [44], together with 38 cell line samples and 1 fetal brain sample from the Harenza dataset [22]. The bidimensional projection showed that Kelly cells sequenced using single-cell resolution were indeed clustering with the bulk Kelly sample, while panobinostat-treated cells were diverging from the control cells but still identifiable within the Kelly lineage.

Differential expression analysis (Appendix A) showed that, including all cells, panobinostat significantly affected the expression of 2708 genes (more than 10% of the entire human transcriptome), of which 2013 were upregulated and 695 downregulated (adjusted *p*-value ≤ 0.01, log_2_ fold change > 1; Figure 2C and Appendix A). The hashtags defining the sample of origin were unsurprisingly amongst the most differentially expressed features defining treated and control cells. Amongst the genes most upregulated by panobinostat, we identified the nucleosome protein-encoding gene H1F0, while amongst the most downregulated, we identified CENPV, encoding for centromere protein V, and BID, encoding for a regulator of apoptosis. The tumor suppressor gene TP53 was amongst those repressed by panobinostat, with an adjusted *p*-value of 2.81 × 10^−11^ and log_2_ fold change of −0.81.

The observed differential expression signature was then compared with another panobinostat treatment experiment involving NBL cells [10]; in this experiment, SK-N-BE(2) cells, another model of MYCN-amplified NBL comparable to our Kelly cells, were treated with 10 nM panobinostat (the same concentration used in the current study) for 6 h (half of the 12 h of our experiment). Gene expression was then measured using the Affymetrix HuGene-2_0-st microarray. Despite these technical differences between the two experiments (different treatment time, different cells, different measuring platform), the effect of the panobinostat at the transcriptional level on MYCN-amplified NBL models was very similar, with SCC = 0.6 (Figure 2D). Several genes were significant and concordant in both datasets, such as the upregulated long noncoding RNA gene MIR7-3HG, the regulator of G protein signaling 16 (RGS16) transcript, or the downregulated genes SRRM2 (encoding for a component of the spliceosome) and the receptor-encoding cholinergic receptor nicotinic alpha 3 subunit (CHRNA3). Unfortunately, microarrays are limited by not possessing probesets for 100% of the transcriptome (hence the lack of data for H1F0 in the SK-N-BE2C dataset).

Pathways globally most altered by panobinostat (Figure 3A and Appendix A) included a significant increase in senescence (Figure 3B), as well as the inhibition of pro-proliferative genes (Figure 3C) [33]. We also noticed a marked overlap with a previous experiment involving HDAC inhibition in multiple myeloma (MM), indicating similar mechanisms in both the MM and NBL contexts [34] (Figure 3D).

Our data suggest that, in parallel with a pleiotropic effect of inhibition of cell growth (Figure 3B,C and Table 1), panobinostat was able to induce mechanisms of neuronal differentiation in a minority of Kelly cells (Figure 3E), an effect that would be lost when analyzing all cells via bulk sequencing (Appendix A).

### 3.3. Master Regulator Analysis to Identify Gene Networks Most Silenced by Panobinostat

In order to understand which transcriptional regulators are most affected by panobinostat, we performed a master regulator analysis (MRA) [46] based on gene regulatory network models [47] extracted from neuroblastoma-specific histological contexts. In order to increase the robustness of the analysis, we adopted different networks derived from patients from the NRC and TARGET cohorts and previously used to define master regulators of specific NBL subtypes [20]. We applied these networks to the panobinostat vs. control signature derived from our experiment, yielding similar results for both networks (Figure 4A,B), including commonly repressed networks, such as BAZ1A, HCFC1, MAZ, MYCN, and ZNF146 (Appendix A). Both analyses showed a high correlation in terms of normalized enrichment scores for all transcription factors, with a marked conservation of downregulated gene networks (Figure 4C). The inhibitory effect of panobinostat on these five master regulators was evident both when analyzing the signature using all panobinostat cells vs. all control cells and when analyzing the dataset as a gradient of progressive panobinostat effect, as shown in Appendix A. We also noticed that many of the master regulators whose networks were most repressed by panobinostat also constituted significant markers of poor prognosis according to survival analyses in NBL datasets and all TCGA cancer datasets (Appendix A and Figure 4D).

In order to determine which transcriptional regulators were most affected by panobinostat—and most associated with tumor aggressiveness and poor prognosis—we integrated evidence based on TF rankings (Appendix A) as follows: TFs whose expression was more repressed by panobinostat in the Shahbazi et al. experiment [10] and in our panobinostat experiment; TFs whose activity (MRA-derived network score) was more downregulated in our experiment using the NRC and TARGET networks; TFs whose association with poor prognosis was more significant in three NBL datasets (Kocak [48], TARGET [20], and NRC [20]); TFs whose association with poor prognosis was more significant in TCGA solid tumors [49]. The combined results are shown in tabular format in Appendix A and indicate that HCFC1, BAZ1A, and ZNF146 were at the center of the gene networks most strongly inhibited by panobinostat and associated with poor cancer prognosis. A Kaplan–Meier curve stratifying NBL patients from the Kocak datasets using a combination of HCFC1, BAZ1A, and ZNF146 is shown in Figure 4D. The TF MAZ, while less significantly associated with prognosis than the top three TFs (Appendix A) and less significantly silenced at the transcriptional level (Appendix A), showed, however, a very similar pattern of network inactivation at the single-cell level (Figure 5).

### 3.4. In Vitro Silencing of Predicted Master Regulators of Panobinostat Response Leads to a Reduction in NBL Cell Growth

Given the effects of panobinostat in reducing cell proliferation (Table 1 and Figure 4), we set out to test if this phenomenon was sustained by the master regulators most consistently repressed by this treatment; namely, the TFs BAZ1A, HCHFC1, MAZ, and ZNF146, which are also associated with poor prognosis in both NBL and pan-cancer (TCGA) datasets. In order to validate the putative proliferative/oncogenic roles of these TFs, we analyzed the growth rate of NBL Kelly cell pools harboring stable and doxycycline-inducible shRNA expression systems. Specifically, we transduced Kelly cells with lentiviral constructs carrying two specific shRNA for each gene analyzed. As reported in Figure 6A, all the shRNA designed effectively knocked-down the expression of all the genes analyzed. Consistently with the bioinformatic prediction, growth rate reduction was recorded after knock-down of BAZ1A, HCHFC1, MAZ, and ZNF146 (Figure 6B), strongly suggesting an oncogenic role for all the putative master regulator genes analyzed.

## 4. Discussion

This study investigated the molecular effects of panobinostat, a pan-HDAC inhibitor, on Kelly cells, a commonly used model for MYCN-amplified NBL [21], using the unprecedented resolution offered by single-cell sequencing [50]. We provide evidence that this technology is now mature enough to fully replace previous transcriptome-wide quantitative technologies, such as microarrays and bulk RNA sequencing (Figure 2). Our experiment recapitulates previous data for Kelly cells from bulk RNA sequencing and provides a novel signature for panobinostat effects highly correlated to that previously observed for another MYCN-amplified NBL model, SK-N-BE(2)C cells (Figure 2D).

Our study shows the magnitude and nature of panobinostat effects on Kelly cells, which deregulated a significant fraction of its transcriptome (14.6%: 2708 significant genes over 18,557 total detected genes). Our article provides a full signature of all genes and their changes, using a dataset composed of more than 2000 individual cells (Appendix A). Pathway enrichment analysis showed that the majority of the observed effects of panobinostat can be summarized as a generalized stopping of proliferation (also highlighted by a shift from the S/G2/M phases to the G1 phase for treated cells (Figure 1F)) and senescence/apoptosis pressure. However, thanks to the potential offered by single-cell resolution, we could also observe a minor—but noticeable—quantity of cells undergoing an early neuronal differentiation process (Figure 4D).

Our analysis also elaborated the transcriptional signature of panobinostat in order to identify which gene networks were most affected by the drug, finding TFs previously unassociated with NBL and confirming the central role of MYCN in NBL tumor sustenance. We successfully recreated the antiproliferative effects of panobinostat by silencing the master regulators identified by our pipeline (HCFC1, BAZ1A, MAZ, and ZNF146), a perturbation that showed a mitigating effect in their role as pro-proliferative and growth-sustaining transcriptional engines in NBL Kelly cells. Targeting these genes could constitute an alternative approach to panobinostat, particularly in patients where the drug exhibits excessive cytotoxic effects or where resistance mechanisms arise [51].

Single-cell sequencing provides a novel tool for the identification of early and/or rare events affecting a small number of cells, which in the future could be leveraged to identify, for example, early mechanisms of drug resistance already arising in vitro. More generally, our study stands as a proof-of-concept that single-cell transcriptomics in combination with ad hoc algorithms developed to extract information from differential expression, such as master regulator analysis, constitute a promising approach to investigating pharmacological perturbations in cell cultures, providing unprecedented resolution and sensitivity in understanding global and peculiar molecular effects.

## Figures and Tables

**Figure 2 genes-13-02240-f002:**
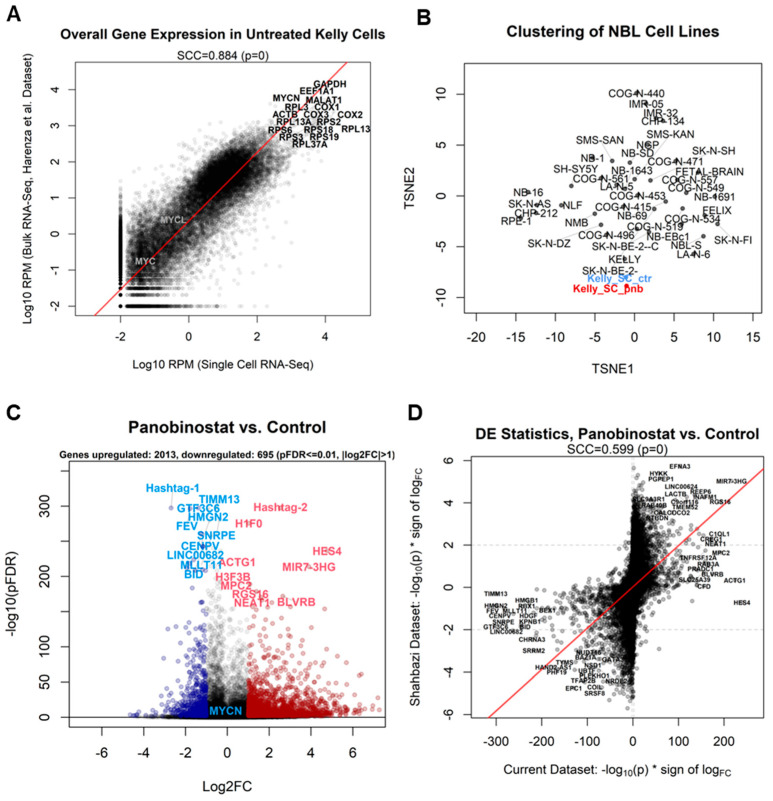
(**A**) Comparison between overall expression of all genes measured in our single-cell experiment (sum of all untreated Kelly cells, x-axis) and the expression of untreated Kelly cells from a bulk RNA-Seq experiment (y-axis) [22]. The Spearman’s correlation coefficient (SCC) is reported. Expression is reported as log_10_ (reads per million mapped reads (RPM)). The most expressed genes in both experiments are reported alongside the MYCN paralogs MYC and MYCL, shown for comparison. (**B**) t-SNE projection of cell lines measured with bulk RNA-Seq from the previously published experiment [22], including pseudobulk counts of all cells in the current single-cell RNA-Seq experiment, divided into control (Kelly_SC_ctr) and treated (Kelly_SC_pnb) groups. Data were VST-normalized [45] prior to t-SNE analysis. (**C**) Volcano plot depicting the differential expression pattern for the contrast panobinostat vs. control, reported as log_2_ fold change (x-axis) vs. −log_10_ (adjusted *p*-value) (y-axis). Genes most upregulated in treated cells are depicted in red, genes most downregulated are shown in blue. MYCN is shown as a comparison, as it is downregulated but with a higher adjusted *p*-value (1.71 × 10^−48^). (**D**) Transcriptome-wide comparative plot of PNB-treated vs. control differential expression signatures in the current experiment (x-axis) vs. the Shahbazi microarray PNB experiment (y-axis). Signatures are expressed as −log_10_ (adjusted *p*-value) multiplied by (*) the sign of log fold change. Genes with the greatest changes in agreement in both experiments are labeled. The Spearman’s correlation coefficient (SCC) is reported.

**Figure 3 genes-13-02240-f003:**
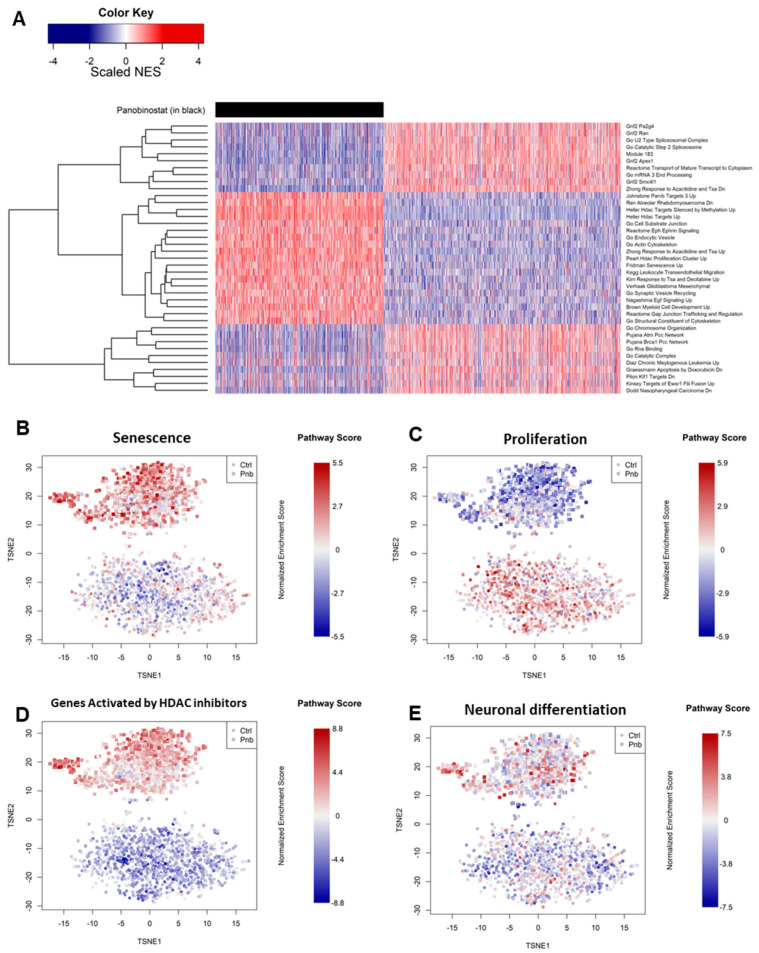
(**A**) Single-cell gene set enrichment analysis (GSEA) reporting the relative normalized enrichment scores (NESs) for pathways (rows) in cells (columns). Cells treated with PNB are grouped to the left and marked in black. Pathways reported are those that most significantly changed in the PNB-treated vs. control contrast. (**B**–**E**) Single-cell gene set enrichment analysis projecting the normalized enrichment score of each selected pathway in each cell of the entire dataset of control (ctrl) and panobinostat-treated (pnb) cells. The selected pathways were defined by the MSigDB database [31] (see Section 2).

**Figure 4 genes-13-02240-f004:**
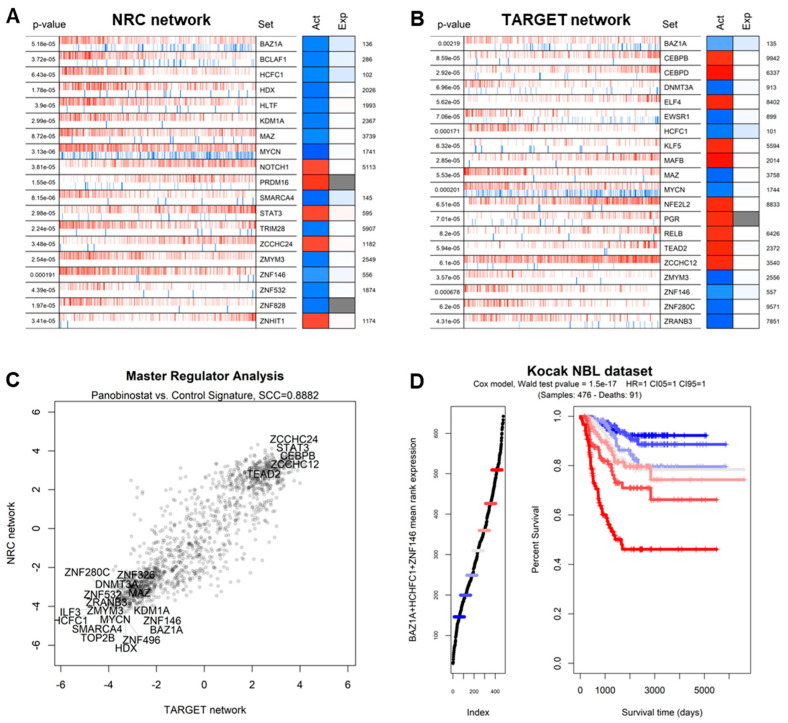
(**A**) Master regulator analysis showing the most significant transcription factors with networks differentially activated in the contrast between PNB-treated Kelly cells and control Kelly cells, graphically depicted as in [20]. In brief, the leftmost column indicates the *p*-value of the observed pattern; the central barcode-like graph indicates the distribution of activated (red bars) and repressed (blue bars) targets of a TF, in positions ranked according to the PNB vs. control signature (leftmost: most downregulated by PNB, rightmost: most upregulated by PNB). The “Act” and “Exp” columns report the normalized enrichment score (NES) as the predicted activity of the whole TF network and the expression level of the TF itself. The number to the right indicates the ranking of the TF in terms of the significance of the transcriptome-wide differential expression analysis. Analysis and plotting were performed with the *viper* R package [35]. The overall network in this panel was derived from the NBL NRC cohort (obtained from [20]). (**B**) Master regulator analysis as in the previous panel with network derived from the TARGET cohort (also obtained from [20]). (**C**) Comparison of master regulator analyses based on the PNB-treated vs. control contrast using two complementary NBL-based networks: TARGET (x-axis) and NRC (y-axis). Candidate MRs with the highest absolute combined NESs are shown as labels. (**D**) Kaplan–Meier curve indicating the association between BAZ1A expression and survival in the Kocak NBL dataset [48]. Patients are divided into groups of increasing BAZ1A expression (left) and shown with regard to survival time vs. percent survival (right). *p*-value calculated with the Wald test [20].

**Figure 5 genes-13-02240-f005:**
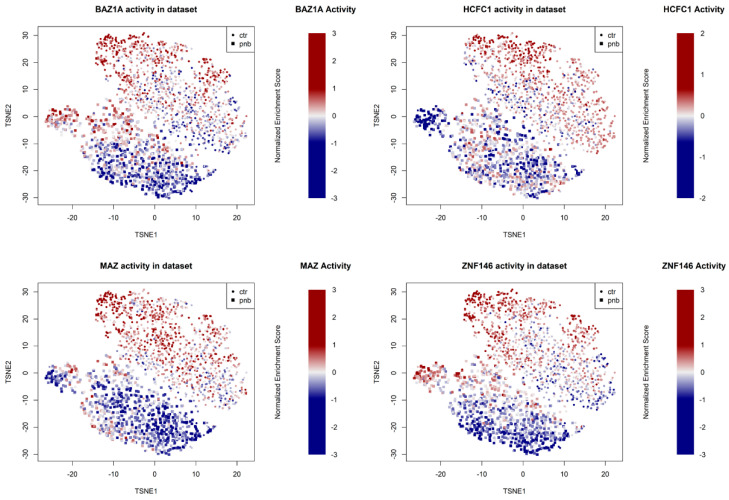
t-SNE projection (calculated after removal of ambiguous multihashtag cells) of the single-cell experiment. Colors are proportional to the normalized enrichment score of the indicated TF network (BAZ1A, HCHFC1, MAZ, and ZNF146) relative to the average of the dataset. Symbols indicate whether the cell was PNB-treated (square) or a control (small circle).

**Figure 6 genes-13-02240-f006:**
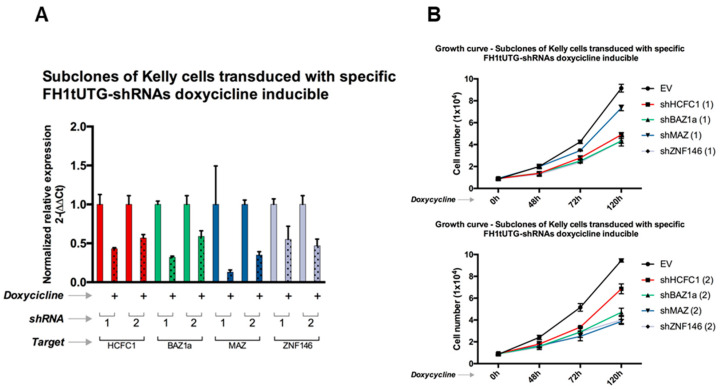
(**A**) Growth curves of Kelly cells stably expressing shRNA for genes HCFC1, BAZ1A, MAZ, and ZNF146, the silencing of which was induced (+) by doxycycline (2 ug/mL) at 0 h, 48 h, 72 h, and 120 h. EV = empty vector. Two different shRNAs were designed for each gene, and the results are shown separately in the top and bottom panels. (**B**) Normalized relative expression for the target gene of each shRNA construct. Error bars indicate the standard error (RT-qPCR was performed in triplicates).

**Table 1 genes-13-02240-t001:** Numbers (and percentages) of control cells and panobinostat-treated Kelly cells at different cell cycle phases (inferred in silico).

	G1	S	G2/M
Control	344 (25.39%)	671 (49.52%)	340 (25.09%)
Panobinostat	447 (47.65%)	340 (36.25%)	151 (16.10%)

## Data Availability

The full results of the differential expression analysis, pathway enrichment analysis, and master regulator analysis are available in the Appendix A. The R code is available on the GitHub repository https://github.com/federicogiorgi/panobinostat (accessed on 22 November 2022), as well as data provided as the RData objects seurat.rda (containing the entire dataset in an R Seurat object) and rawcounts.rda (raw count gene expression matrix, with genes as rows and cells as columns).

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
