# Peer review of "Single-Cell Sequencing Identifies Master Regulators Affected by Panobinostat in Neuroblastoma Cells"

_genes, 2022, doi:10.3390/genes13122240_

Round 1

Reviewer 1 Report

This paper described the single-cell sequencing data in neuroblastoma cells treated by Panobinostat. The methodology of this paper is interesting but there are several critical problems as follows:

1. First of all, why the authors used single-cell sequencing in this study? The results seem to be similar to that of bulk data.

2. As you know, many neuroblastoma cell lines have been established. The authors used one cell line, Kelly cells. If you analyze neuroblastoma cell lines, several diffrent cell lines with or wihtout MYC amplification should be used.

3. In this study, the Panobinostat concentration was 10nM and treatment time was 12 hours.  How did the authors decide this appropriate condition?  The effectiveness of Panobinostat  in neuroblastoma is unclear. The treatment condition is critical and single cell seq will be better to be analyzed  in several timings in the treatment.

4. Using Master Regulator Analysis, the authors identified BAZ1A, HCFC1, MAZ, and ZNF146 as the transcriptional regulators most significantly repressed by Panobinostat. This process is unclear.

5. In the method, the authors describe Dox-inducible shRNA expression cell lines. This result was not described in this paper.

6. Several abbreviations such as TF and TSNE were unclear. 

7. The gene names described in Figs. 2 and 3 were unable to be read.

Author Response

The reply to reviewer is provided in the attached word document responseToReviewers.docx

Reviewer 2 Report

Single-Cell sequencing identifies Master Regulators affected by Panobinostat in Neuroblastoma cells

In this manuscript Giorgio et al., authors reported transcriptomic profile of neuroblastoma under drug treatment, Panobinostat. Among other drugs, Panobinostat has been studied a lot pre-clinically and clinically to treat tumor but still missing its mechanistic and cellular role. Author has described well their aim using scRNAseq approach. In this study, author uses mainly bio-informatics approach i.e., single cell RNA sequencing to answer their research goal.

The manuscript proposes a unique finding but need major revision and explanation.

Comments

Major Revision:

1.    Author did not describe the how many sets of experiments (control vs treatment) were submitted for sequencing. I assume that at least three-five sets need to be submitted in order to conclude the results.

2.   In their clustering method, Author did not describe what parameter have been used.

3.    Author did not mention what cluster type difference between control and treatment.

4.    It is not clear that why author mentioned about hastagging concept in their scRNAseq data analysis.

5.    I would be interesting to compare most variable cluster between control and treatment. For example, Heatmap to show the expression top differentially expressed genes between control and treatment [for each cluster].

6.    Author described about senescence and pro-proliferative genes as a unique pathway in their dataset. In addition to control vs treatment heatmap, I would also prefer having a figure for pathways analysis comparison for most varied clusters (average expression based, heatmap).

7.   Author may add a vinplot to show enrichment score for senescence, proliferation and senescence for each cluster from scRNAseq data (to see which cluster has more chance of such events).

8.   It would be nice to have venn-diagram to show how many genes are common shared between control and treatment. List unique genes in the treatment if possible.

9.   Author have not looked at whether senescence and pro-proliferative signals are coming from same cell or there are two sub-population of cells/cluster to drive the signals.

10.  Author already knows the common repressor gene in their pathway, I would suggest taking gene sets and apply U cell package to see if these gene sets are fitting to which clusters.

Package: GitHub - carmonalab/UCell: An R package for single-cell gene signature scoring

11.    It is minor, but did author check the level of tumor suppressor genes (p53 or p21). As, these cells are halted in G1 phase, there might be a chance they have high level of these candidate to inhibit proliferation and induce senescence.

12.    Manuscript needs more background explanation in material and method section, but I liked the content of research.

13.    Author should provide processing and figure code as well for the update version of manuscript to review and replicate at our end. Sometimes, methods are not very descriptive to understand in the article.

14.    I am assuming that author would also submit their data and code to Github repository.

Author Response

(The authors gave the same response as above.)

Reviewer 3 Report

The manuscript by Milazzo et al presents an interesting study investigating the molecular effects of Panobinostat, a pan- histone deacetylase inhibitor, on a cell model of neuroblastoma, Kelly cells, by using single-cell sequencing.

Authors analyze transcriptional effects of treatment, identifying up- and down regulated genes as well as increased expression effects on master transcriptional regulators. Additional models showing the same MYCN amplification as Kelly cells were also used to compared results.

Master regulators of transcription identified were knocked down in Kelly cells, and reduction in cells growth was observed.

The article is well written, and adds to the power of these single-cell sequencing techniques in identifying drug’s effects. Conclusions are supported by results.

Minor

The TSNE analysis is presented when showing clustering of genes differentially expressed. This t-distributed stochastic neighbor embedding techniques is usually represented as t-SNE

line 349, spell out KD

Author Response

(The authors gave the same response as above.)

Round 2

Reviewer 2 Report

Now, Revised manuscript is in much better shape. I do not have further comment. Authors may check some minor spelling mistakes or text formatting.